# Hydra-MDP++: Advancing End-to-End Driving via Expert-Guided Hydra-Distillation

## Abstract

We introduce Hydra-MDP++ , a novel end-to-end autonomous driving framework that integrates rule-based and neural planners by learning from human demonstrations and distilling knowledge from rule-based experts. We propose a teacher-student knowledge distillation framework with a multi-head student decoder that integrates feedback from rule-based expert teachers. The student model achieves state-of-the-art performance on the NAVSIM benchmark with a tiny image encoder. Moreover, to address limitations in existing evaluation metrics, we expand the teacher model to include traffic light compliance, lane-keeping ability, and extended comfort. This is intended to ensure a more robust decision synthesis in driving. Hydra-MDP++ demonstrates robust and efficient performance across diverse driving scenarios, achieving a 91.0% drive score on NAVSIM by simply scaling the image encoder. Our work contributes to developing more reliable and adaptable autonomous driving systems that combine the strengths of rule-based and neural planning approaches.

## 1 INTRODUCTION

Developing reliable and robust motion planning systems remains a critical challenge in the rapidly evolving field of autonomous driving. Traditional **rule-based planners** (Treiber et al., 2000; Thrun et al., 2006; Bacha et al., 2008; Leonard et al., 2008; Fan et al., 2018; Dauner et al., 2023) have long been a cornerstone of autonomous driving systems. These planners rely on predefined rules and heuristics to make decisions, offering **transparency and interpretability**. They excel in handling well-defined scenarios and can be easily adjusted to comply with traffic regulations. However, they often struggle with the unpredictability and complexity of real-world driving situations.

On the other hand, there is a well-established history of using neural networks for vision-based steering control and autonomous driving (Pomerleau, 1988; Lecun et al., 2004; Bojarski et al., 2016). **Neural planners** (Bansal et al., 2018; Qureshi et al., 2019) and their applications in end-to-end autonomous driving (Codevilla et al., 2018; Zeng et al., 2019; Wu et al., 2022; Hu et al., 2022; 2023; Jiang et al., 2023; Wang et al., 2023b; Chen et al., 2024) have gained significant attention in recent years. These data-driven approaches can learn from vast amounts of driving data, potentially capturing nuances of driving behavior that are difficult to encode in rule-based systems. End-to-end autonomous driving are noteworthy for their ability to process all available image features directly through to the planning stage, enabling them to capture and respond to subtle cues and complex interactions in driving scenarios. This capability allows them to potentially make more nuanced and context-aware decisions, enhancing the overall performance and safety of autonomous vehicles. While these planners promise **adaptability** to more diverse scenarios, they can sometimes be opaque in their decision-making process.

Rule-based and neural planners have long been viewed as occupying the opposite ends of the autonomous driving spectrum. However, we argue that this perceived dichotomy may be overstated. The gap between these approaches can be bridged by expanding the capabilities of neural planners beyond mere imitation of **human demonstrations** to knowledge distillation from **interpretable rule-based experts**. While imitation learning excels at replicating human actions in specific scenarios, it often overlooks critical safety considerations (Dauner et al., 2023; Li et al., 2023). In contrast, those rule-based experts focus on the key components essential to safe and efficient driving: adherence to traffic rules, collision avoidance, and driving comfort.

Figure 1: **Comparisons between three paradigms for autonomous driving solutions.**

Therefore, we propose a novel end-to-end autonomous driving framework that learns to incorporate both human demonstrations and expert-guided decision analysis, namely Hydra-MDP++ . This framework utilizes a teacher-student knowledge distillation (KD) architecture to capture the essence of human-like driving behavior. The student model generates diverse trajectory candidates, while teacher models, derived from human demonstrations and rule-based systems, validate these proposals based on various aspects of expert driving knowledge. We implement this multi-target validation using a multi-head decoder, enabling the integration of feedback from specialized teachers that represent different components of safe and efficient driving, namely Hydra-Distillation. We compare our proposed approach, the previous neural planners, and rule-base planners in Fig. 1.

We evaluate our approach on NAVSIM, a data-driven non-reactive autonomous vehicle simulation and benchmarking tool. NAVSIM provides a middle ground between open-loop and closed-loop evaluations using large datasets combined with a non-reactive simulator. It gathers simulation-based metrics such as progress and time-to-collision by unrolling bird's eye view abstractions of test scenes for a short simulation horizon. This non-reactive simulation allows for efficient, open-loop metric computation while aligning better with closed-loop evaluations than traditional displacement errors. As an extension of the NAVSIM challenge-winning solution Hydra-MDP (Li et al., 2024), Hydra-MDP++ leverages a simple encoder-decoder architecture. For the encoder, it simply uses classic pretrained vision encoders, such as ResNet-34 (He et al., 2016) or VoVNet-99 (Lee & Park, 2020). The decoder employs a lightweight and simple transformer network. Hydra-MDP++ achieved state-of-the-art performance on NAVSIM using only a lightweight ResNet-34 network without additional complex components. Hydra-MDP++ can easily outperform and achieve a 91.0% PDM Score by simply scaling the image encoder.

Moreover, we found that the NAVSIM-derived teachers do not sufficiently capture the full spectrum of driving decision-making, potentially leading to unsafe behaviors. To address this, we expand the original teacher by incorporating traffic light compliance (TL), lane-keeping ability (LK), and extended comfort (EC).

We summarize our contributions as follows:

1. We introduce Hydra-MDP++ , a novel end-to-end autonomous driving framework that incorporates both human demonstrations and rule-based experts.

2. Our proposed approach achieves top performance on NAVSIM using only a lightweight ResNet-34 network. Hydra-MDP++ achieves a 91.0% drive score by scaling the image encoder to V2-99.

3. We address the issues in the NAVSIM-derived teachers by incorporating traffic light compliance (TL), lane-keeping ability (LK), and extended comfort (EC) teachers to reflect better-driving decision-making.

## 2 RELATED WORK

### 2.1 END-TO-END AUTONOMOUS DRIVING

End-to-end autonomous driving streamlines the entire stack from perception to planning into a single optimizable network. This eliminates the need for manually designing intermediate representations. Following pioneering work (Lyu et al., 2019; Bojarski et al., 2016; Kendall et al., 2019), a diverse landscape of end-to-end models has emerged. For example, many end-to-end approaches focus on closed-loop simulators that use single-frame cameras, LiDAR point clouds, or a combination of both to mimic expert behaviour in autonomous driving, namely imitation learning (IL). Expert behaviour typically takes two forms, trajectories and control actions. Transfuser and its variants (Prakash et al., 2021; Chitta et al., 2022) use a simple GRU to auto-regressively predict waypoints for autonomous driving. LAV (Chen & Krähenbühl, 2022) adopts a temporal GRU module to further refine the trajectory. UniAD (Hu et al., 2023) first integrates perception, prediction, and planning into a unified transfomer network. VAD (Jiang et al., 2023) models the driving scene using vectorized representations for efficiency. The recent work PARA-Drive (Weng et al., 2024) implements a fully parallel end-to-end autonomous driving architecture, surpassing the performance of VAD and tripling the processing speed. While they all perform impressively on the NAVSIM benchmark (Contributors, 2024; Dauner et al., 2024), we found that these methods still do not fully mimic human behaviour and score low on certain metrics. This suggests that imitation learning still has limitations.

### 2.2 RULE-BASED AUTONOMOUS DRIVING

Rule-based planners offer a structured, interpretable decision-making framework (Treiber et al., 2000; Thrun et al., 2006; Bacha et al., 2008; Leonard et al., 2008; Fan et al., 2018; Dauner et al., 2023). They have been foundational in ensuring safety and predictability by encoding explicit traffic rules and heuristics into the driving system (e.g., apply a hard brake when an object is straight ahead). One widely used framework is the Intelligent Driver Model (IDM (Treiber et al., 2000)), which governs vehicle acceleration and braking behavior based on relative distances and speeds, enabling safe and efficient car-following in various traffic conditions. Extensions of IDM (Fan et al., 2018) further build on rule-based principles by integrating explicit traffic laws and driving heuristics to guide decision-making in complex environments like urban traffic, leveraging a modular architecture for tasks such as lane-changing and intersection handling. The recent study (Dauner et al., 2023) proposes a rule-based planner named PDM-Planner. It assesses the current state of closed-loop planning in the field, revealing the limitations of learning-based methods in complex real-world scenarios and the value of simple rule-based priors such as centerline-following and collision avoidance.

### 2.3 CLOSED-LOOP BENCHMARKING WITH SIMULATION

Closed-loop benchmarking with simulation is essential for evaluating autonomous driving systems by measuring key aspects like safety, rule compliance, and comfort. Tools such as CARLA (Dosovitskiy et al., 2017) and Metadrive (Li et al., 2022) focus on sensor-based simulations, mimicking the real world through virtual cameras and LiDAR. In contrast, platforms like nuPlan (Karnchanachari et al., 2024) and Waymax (Gulino et al., 2024) utilize data-driven approaches to simulate urban environments with real-world data, providing a more dynamic and realistic evaluation of autonomous agents. Despite advancements, replicating accurate traffic behaviors and realistic sensor data remains challenging. Achieving high fidelity in environmental interactions and vehicle dynamics is crucial for ensuring reliability in autonomous systems. Existing data-driven sensor simulations attempt to bridge this gap by adapting real sensor data for new driving scenarios, but they still fall short in rendering quality. For instance, many simulations depend on graphical-based rendering to replicate sensor input like LiDAR, often resulting in discrepancies when compared to real-world conditions. Simulators integrating more realistic sensor behavior, like CARLA and Metadrive, still face hurdles in domain gaps related to visual fidelity and sensor precision. The latest NAVSIM benchmark (Contributors, 2024; Dauner et al., 2024) addresses some of these challenges by evaluating planners with extended metrics, such as driving consistency and safety, that better reflect real-world driving performance. However, it assumes a non-reactive environment, where agents do not influence their surroundings over a short horizon, simplifying real-world dynamics in its assess-

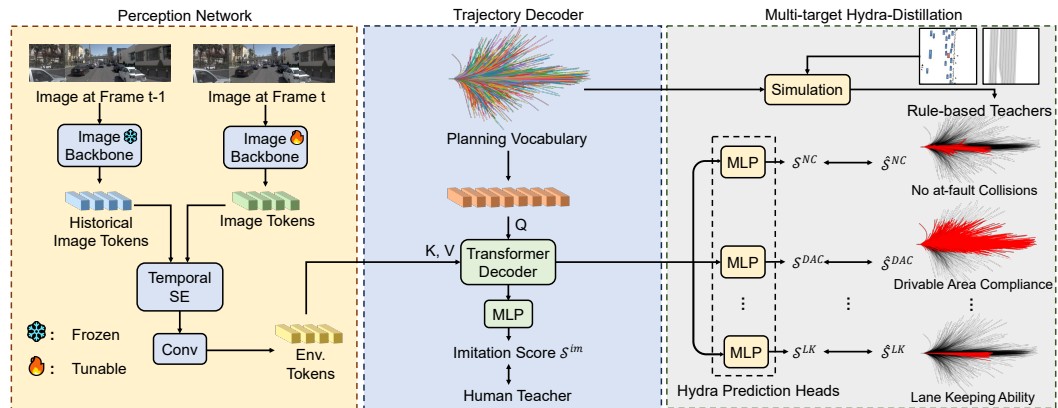

Figure 2: **The Overall Architecture of Hydra-MDP++ .**

ment framework. While NAVSIM provides more accurate simulation-based evaluation, closing the gap between simulated and real-world conditions remains an ongoing area of research.

# 3 METHODOLOGY

## 3.1 PRELIMINARIES

**Imitation-based End-to-end Neural planners.** Neural planners utilize deep learning to predict driving trajectories or control commands directly from raw sensory inputs, such as camera images and LiDAR data. They can automatically extract relevant patterns and adapt to complex environments after being trained on extensive datasets. Specifically, the neural planner learns to replicate human driving behavior by predicting a trajectory $T$ or a sequence of control actions $(a_1, a_2, ..., a_T)$, supervised by human demonstrations. Nevertheless, these approaches may lack interpretability seen in rule-based systems.

**Rule-based planners.** On the other hand, rule-based planners adhere to predefined rules and expert knowledge to ensure safe and efficient driving. As the state-of-the-art planner on the nuPlan dataset (Caesar et al., 2021b), the PDM-Planner (Dauner et al., 2023) integrates the Intelligent Driver Model (IDM Treiber et al. (2000)) with various hyperparameters to enhance performance. This approach evaluates multiple planning proposals through a comprehensive metric known as the PDM Score (PDMS):

$$\text{PDMS} = \underbrace{\left( \prod_{m \in \{\text{NC,DAC}\}} \text{S}^m \right)}_{\text{penalties}} \times \underbrace{\left( \frac{\sum_{w \in \{\text{EP,TTC,C}\}} \text{weight}_w \times \text{S}^w}{\sum_{w \in \{\text{EP,TTC,C}\}} \text{weight}_w} \right)}_{\text{weighted average}}, \quad (1)$$

which addresses various aspects of driving performance, including safety, comfort, and progress. In our framework, we utilize the metric system of the PDM-Planner as teachers, which broaden the learning objective of the planner.

## 3.2 OVERALL FRAMEWORK

As shown in Fig. 2, Hydra-MDP++ consists of two networks: a **Perception Network** and a **Trajectory Decoder**.

**Perception Network.** Our perception network consists of an image backbone and a temporal Squeeze-and-Excitation (SE) network for temporal fusion. The temporal SE network builds on the classic SE network (Hu et al., 2018), which performs channel-wise attention, but in our case, it is adapted to operate across the temporal dimension. This mechanism aggregates both historical

$(F_{img}^{pre})$ and current $(F_{img}^{cur})$ image features through a squeeze operation, compressing temporal information. The excitation step then learns weights that highlight important temporal dependencies, improving the model's ability to adapt to dynamic environments. The environmental tokens $F_{env}$, encoding rich temporal and semantic information, are computed as follows:

$$F_{env} = Conv(TemporalSE(Concat(F_{img}^{pre}, F_{img}^{cur}))). \tag{2}$$

By applying attention across time, the Temporal SE module enhances the performance when processing sequential data. We detach the gradient of historical tokens for faster convergence (Wang et al., 2023a; Yuan et al., 2024b).

**Trajectory Decoder.** Based on these environmental tokens, many end-to-end planners primarily regress to a single target trajectory (Jiang et al., 2023; Hu et al., 2023; Wu et al., 2022), which only imitates human behavior and fails to address the uncertainty in planning (Chen et al., 2024). This approach is limited by its reliance on extrapolating historical ego status, leaving a significant action space unexplored. On the other hand, a discretized action space (Philion & Fidler, 2020; Phan-Minh et al., 2020; Chen et al., 2024) not only helps us avoid these problems but also enables the generation of the ground-truth metric data for expert-guided knowledge distillation offline, as long as the action space is fixed during training. To construct the action space, we first sample 700K trajectories randomly from the original nuPlan database (Caesar et al., 2021a). Each trajectory $T_i (i = 1, ..., k)$ consists of 40 timestamps of $(x, y, heading)$, corresponding to the desired 10Hz frequency and a 4-second future horizon. The planning vocabulary $\mathcal{V}_k$ is formed as K-means clustering centers of the 700K trajectories, where $k$ denotes the size of the vocabulary. $\mathcal{V}_k$ is then embedded as $k$ latent queries with an MLP, sent into layers of transformer encoders (Vaswani et al., 2017), and added to the ego status $E$:

$$\mathcal{V}'_k = Transformer(Q, K, V = Mlp(\mathcal{V}_k)) + E. \tag{3}$$

To incorporate environmental clues in $F_{env}$, transformer decoders are leveraged:

$$\mathcal{V}''_k = Transformer(Q = \mathcal{V}'_k, K, V = F_{env}). \tag{4}$$

### 3.3 LEARNING AND INFERENCE

The learning process of this architecture consists of two key elements: **Imitation Learning** and **Expert-guided Hydra-Distillation**, as illustrated in Fig. 2. Through Imitation Learning, the model learns from human demonstrations. Expert-guided Hydra-Distillation provides additional guidance from a rule-based expert, ensuring the model to adhere to driving rules and safety standards. This approach combines the flexibility of learning from demonstrations with the reliability of rule-based corrections, leading to improved driving performance in real-world scenarios.

**Imitation Learning.** With a classification-based trajectory decoder, the primary objective is to estimate the confidence of each trajectory. To reward trajectory proposals that are close to human driving behaviors, we implement a distance-based cross-entropy loss to imitate the log-replay trajectory $\hat{T}$ derived from humans:

$$\mathcal{L}_{im} = -\sum_{i=1}^{k} y_i \log(\mathcal{S}_i^{im}), \tag{5}$$

where $\mathcal{S}_i^{im}$ is the $i$-th softmax score of $\mathcal{V}''_k$, and $y_i$ is the imitation target produced by L2 distances between log-replays and the vocabulary. Softmax is applied on L2 distances to produce a probability distribution:

$$y_i = \frac{e^{-(\hat{T}-T_i)^2}}{\sum_{j=1}^{k} e^{-(\hat{T}-T_j)^2}}. \tag{6}$$

**Expert-guided Hydra-Distillation.** Though the imitation target provides certain clues for the planner, it is insufficient for the model to associate the planning decision with the driving environment, leading to failures such as collisions and leaving drivable areas (Li et al., 2023). Therefore, to improve the closed-loop performance of our end-to-end planner, we propose Expert-guided Hydra-Distillation, a learning strategy that aligns the planner with simulation-based metrics in NAVSIM.

The distillation process expands the learning target through two steps: (1) running offline simulations (Dauner et al., 2023) of the planning vocabulary $\mathcal{V}_k$ for the entire training dataset; (2) introducing supervision from simulation scores for each trajectory in $\mathcal{V}_k$ during the training process. For

a given scenario, step 1 generates ground truth simulation scores $\{\hat{\mathcal{S}}_i^m | i = 1, ..., k\}_{m=1}^{|M|}$ for each metric $m \in M$ and the $i$-th trajectory, where $M$ represents the set of metrics metioned in Sec. 3.1 and Sec. 3.4, excluding extended comfort metric. For score predictions, latent vectors $\mathcal{V}_k''$ are processed with a set of Hydra Prediction Heads, yielding predicted scores $\{\mathcal{S}_i^m | i = 1, ..., k\}_{m=1}^{|M|}$. With a binary cross-entropy loss, we distill rule-based driving knowledge into the end-to-end planner:

$$\mathcal{L}_{kd} = -\sum_{m,i} \hat{\mathcal{S}}_i^m \log \mathcal{S}_i^m + (1 - \hat{\mathcal{S}}_i^m) \log(1 - \mathcal{S}_i^m). \tag{7}$$

For a trajectory $T_i$, its distillation loss of each sub-score acts as a learned cost value, measuring the violation of particular traffic rules associated with that metric. The overall loss $L$ can be expressed as follows:

$$\mathcal{L} = \mathcal{L}_{im} + \mathcal{L}_{kd}. \tag{8}$$

**Inference.** Given the predicted imitation scores $\{\mathcal{S}_i^{im} | i = 1, ..., k\}$ and metric sub-scores $\{\mathcal{S}_i^m | i = 1, ..., k\}_{m=1}^{|M|}$, we calculate an assembled cost measuring the likelihood of each trajectory being selected in the given scenario as follows:

$$\tilde{f}(T_i, O) = -(k_{im} \log \mathcal{S}_i^{im} + \sum_{m \in M_{penalties}} k_m \log \mathcal{S}_i^m + k_w \log \sum_{w \in M_{weighted}} \text{weight}_w \mathcal{S}_i^w), \tag{9}$$

where $\{k_{im}, k_m, k_w\}$ represent confidence weighting parameters to mitigate the imperfect fitting of different teachers. $M_{penalties}$ and $M_{weighted}$ represent penalty and weighted metrics used in the PDM-Planner (see Eq. 1) and the extended metrics we propose in Sec. 3.4. The optimal combination of weights is obtained via grid search. Finally, the trajectory with the lowest overall cost is chosen.

## 3.4 EXTENDED RULE-BASED TEACHERS

Hydra-MDP++ exhibits strong performance on the NAVSIM benchmark. Nevertheless, we observe certain issues in planned trajectories of the model, which are not perfectly covered by existing metrics used by NAVSIM. These issues include traffic rule violations, deviation from the centerline, and inconsistent predictions between consecutive frames, which can result in oscillation. In light of these observations, we expand the original teacher by incorporating Traffic Lights Compliance (TL), Lane Keeping Ability (LK), and Extended Comfort (EC). Furthermore, our framework is capable of integrating additional rule-based teachers in the event that new rules are designed.

**Traffic Lights Compliance.** It is essential for all vehicles to follow traffic signals, represented by the metric $S^{TL}$. This metric evaluates whether a vehicle runs a red light. Specifically, for the upcoming four seconds, if the vehicle crosses a crosswalk while the light is red, it will be flagged for running the red light. In such an event, $S^{TL}$ is set to 0. However, if the vehicle complies and avoids crossing during the red light, $S^{TL}$ is assigned a value of 1.

**Driving Direction Compliance.** The $S^{DDC}$ metric is employed to determine whether the trajectory of the vehicle between two consecutive time steps remains aligned with the centreline's positive direction, within an allowable distance deviation of $\tau_D$. In the context of time steps $i$ and $i + 1$, the vehicle's positions are defined as $(x_i, y_i)$ and $(x_{i+1}, y_{i+1})$, respectively. The closest lane segment, $v_j$, is then identified, and the projections of these two positions onto the positive direction of $v_j$ are calculated. The distance between the two projected points is subsequently defined as $d_i^p$. The subscore $S^{DDC} = 1$ if, for every time steps i, the condition: $d_i^p \le \tau_D$ holds.

**Lane Keeping Ability.** The lane keeping subscore $S^{LK}$ assesses a vehicle's ability to stay within a lateral deviation limit $\tau_D$ from the lane. This subscore reflects how effectively the vehicle can maintain its intended path during navigation. At each time step $i$, we calculate the minimum perpendicular distance $d_i$ between the ego vehicle $(x_i, y_i)$ and nearby lane segments $v_j$:

$$d_i = \min_{v_j \in m} \{d((x_i, y_i), v_j)\}. \tag{10}$$

The subscore $S^{LK} = 1$ if, for every time steps $i$, the condition: $d_i \le \tau_D$ holds.

**Extended Comfort.** We find that the previous metrics were insufficient in addressing inconsistencies arising from the model's own predictions. For example, if the trajectory predicted in the

previous frame shifts to the left while the current frame's prediction shifts to the right, this can cause vehicle to oscillate, negatively affecting passenger comfort. Accordingly, the extended comfort sub-score $S^{EC}$ is calculated by comparing the discrepancies in acceleration, jerk, yaw rate, and yaw acceleration between the projected trajectories of the preceding and current frames with respect to predefined thresholds $\tau_A$, $\tau_J$, $\tau_Y^R$ and $\tau_Y^A$. The discrepancies are calculated as follows:

$$d_A = \sqrt{\frac{1}{T}\sum_{t=1}^{T}\left(a_{\text{current},t} - a_{\text{preceding},t}\right)^2}, \quad d_J = \sqrt{\frac{1}{T}\sum_{t=1}^{T}\left(j_{\text{current},t} - j_{\text{preceding},t}\right)^2}, \quad (11)$$

$$d_Y^R = \sqrt{\frac{1}{T}\sum_{t=1}^{T}\left(y_{\text{current},t}^r - y_{\text{preceding},t}^r\right)^2}, \quad d_Y^A = \sqrt{\frac{1}{T}\sum_{t=1}^{T}\left(y_{\text{current},t}^a - y_{\text{preceding},t}^a\right)^2}. \quad (12)$$

The subscore $S^{EC} = 1$ if the condition $d_A \leq \tau_A$, $d_J \leq \tau_J$, $d_Y^R \leq \tau_Y^R$, and $d_Y^A \leq \tau_Y^A$ holds.

In light of the aforementioned four new metrics, the Extended PDM Score can be described as follows:

$$\text{EPDMS} = \underbrace{\left(\prod_{m\in\{\text{NC,DAC,DDC,TL}\}} S^m\right)}_{\text{penalties}} \times \underbrace{\left(\frac{\sum_{w\in\{\text{EP,TTC,C,LK,EC}\}} \text{weight}_w \times S^w}{\sum_{w\in\{\text{EP,TTC,C,LK,EC}\}} \text{weight}_w}\right)}_{\text{weighted average}}. \quad (13)$$

## 4 EXPERIMENTS

### 4.1 DATASET AND METRICS

**Dataset.** The NAVSIM dataset builds on the existing OpenScene (Contributors, 2023) dataset, a compact version of nuPlan (Caesar et al., 2021b) with only relevant annotations and sensor data sampled at 2 Hz. The dataset primarily focuses on scenarios involving changes in intention, where the ego vehicle's historical data cannot be extrapolated into a future plan. The dataset provides annotated 2D high-definition maps with semantic categories and 3D bounding boxes for objects. The dataset is split into two parts: Navtrain and Navtest, which respectively contain 1192 and 136 scenarios for training/validation and testing.

**Metrics.** For NAVSIM dataset, we evaluate our models based on the PDM score (PDMS) and the Extended PDM Score (EPDMS), which can be formulated as follows:

$$PDM_{score} = NC \times DAC \times \frac{(5\times TTC + 2\times C + 5\times EP)}{12}, \quad (14)$$

$$EPDM_{score} = NC \times DAC \times DDC \times TL \times \frac{(5\times TTC + 2\times C + 5\times EP + 5\times LK + 5\times EC)}{22}, \quad (15)$$

where sub-metrics $NC$, $DAC$, $TTC$, $C$, $EP$, $DDC$, $TL$, $LK$ and $EC$ correspond to the No at-fault Collision, Drivable Area Compliance, Time-to-Collision, Comfort, Ego Progress, Drivable Area Compliance, Traffic Lights Compliance, Lane Keeping Ability and Extended Comfort. In regard to the PDM score, we calculate EPDMS following the NAVSIM Benchmark (Contributors, 2024; Dauner et al., 2024) by expanding the original penalty and weighted terms with our proposed metrics.

### 4.2 IMPLEMENTATION DETAILS

We train our models on the Navtrain split (Contributors, 2024) using 8 NVIDIA V100 GPUs, with a total batch size of 256 across 20 epochs. The learning rate and weight decay are set to $1 \times 10^{-4}$ and 0.0, following the official baseline using the AdamW (Loshchilov, 2017) optimizer. For images, the front-view image is concatenated with the center-cropped front-left-view and front-right-view images, yielding an input resolution of $256 \times 1024$ by default. ResNet34 is applied for feature extraction unless otherwise specified. Although the dataset provides four past frames, our model only utilizes the two most recent ones. No data or test-time augmentations are used. Our input data includes the current status of the ego vehicle, such as velocity, acceleration, and driving commands from the navigation module, including turning, lane changing, and following. The final output is a 40-waypoint trajectory over 4 seconds, sampled at 10 Hz, with each waypoint defined by x, y, and

heading coordinates. In the case of extended rule-based metrics, the value of $\tau_D$ was set at $0.5m$ for both Driving Direction Compliance (DDC) and Lane Keeping Ability (LK). Besides, thresholds are set to $\tau_A = 0.7m/s^2$, $\tau_J = 0.5m/s^3$, $\tau_A = 0.7m/s^2$, $\tau_Y^R = 0.1rad/s$, and $\tau_Y^A = 0.1rad/s^2$ in Extended Comfort (EC).

| Method | Inputs | Img. Backbone | Params (MB) ↓ | NC ↑ | DAC ↑ | EP ↑ | TTC ↑ | C ↑ | PDMS ↑ |
|---|---|---|---|---|---|---|---|---|---|
| PDM-closed (Dauner et al., 2023)* | GT Perception | - | - | 94.6 | 99.8 | 89.9 | 86.9 | 99.9 | 89.1 |
| Transfuser (Chitta et al., 2022) | Img.+LiDAR | ResNet34 | 56.0 | 97.7 | 92.8 | 79.2 | 92.8 | 100 | 84.0 |
| UniAD (Hu et al., 2023) | Img. | ResNet34 | - | 97.8 | 91.9 | 78.8 | 92.9 | 100 | 83.4 |
| PARA-Drive (Weng et al., 2024) | Img. | ResNet34 | - | 97.9 | 92.4 | 79.3 | 93.0 | 99.8 | 84.0 |
| VADv2 (Chen et al., 2024)† | Img.+LiDAR | ResNet34 | 60.5 | 97.9 | 91.7 | 77.6 | 92.9 | 100 | 83.0 |
| DRAMA (Yuan et al., 2024a) | Img.+LiDAR | ResNet34 | 50.6 | 98.0 | 93.1 | 80.1 | 94.8 | 100 | 85.5 |
| Hydra-MDP++ (Ours) | Img. | ResNet34 | **40.6** | 97.6 | 96.0 | 80.4 | 93.1 | 100 | 86.6 |
| Hydra-MDP++ (Ours) | Img. | V2-99 | 89.0 | **98.6** | **98.6** | **85.7** | **95.1** | **100** | **91.0** |

Table 1: **Performance on the Navtest Benchmark with original metrics.** The table displays the percentages of the No at-fault Collision (NC), Drivable Area Compliance (DAC), Time-to-Collision (TTC), Comfort (C), and Ego Progress (EP) subscores, as well as the PDM Score (PDMS). *PDM-Closed is provided for reference only due to limitations in the brake implementation, which potentially leads to more collisions. † VADv2 is our implementation based on Transfuser, incorporating a classification-based trajectory decoder.

| Method | Backbone | NC ↑ | DAC ↑ | EP ↑ | TTC ↑ | C ↑ | TL ↑ | DDC ↑ | LK ↑ | EC ↑ | EPDMS ↑ |
|---|---|---|---|---|---|---|---|---|---|---|---|
| PDM-closed (Dauner et al., 2023)* | - | 94.6 | 99.8 | 89.9 | 86.9 | 99.9 | 100 | 98.7 | 66.7 | 98.0 | 82.8 |
| Transfuser (Chitta et al., 2022) | ResNet34 | 97.7 | 92.8 | 78.4 | 93.0 | 100 | 99.9 | 98.3 | 67.6 | 95.3 | 77.8 |
| VADv2 (Chen et al., 2024)† | ResNet34 | 97.3 | 91.7 | 77.6 | 92.7 | 100 | 99.9 | 98.2 | 66.0 | 97.4 | 76.6 |
| Hydra-MDP++(Ours) | ResNet34 | 97.9 | 96.5 | 79.2 | 93.4 | 100 | 100 | 98.9 | 67.2 | **97.7** | 80.6 |
| Hydra-MDP++(Ours) | V2-99 | **98.8** | **97.8** | **84.0** | **95.3** | **100** | **100** | **99.1** | **70.1** | 96.8 | **84.1** |

Table 2: **Performance on the Navtest Benchmark with extended metrics.** The table shows percentages of the original metrics and our proposed metrics Traffic Lights Compliance (TL), Driving Direction Compliance (DDC), Lane Keeping Ability (LK), Extended Comfort (EC), and the Extended PDM score (EPDMS). * and † have the same meaning as in the previous table.

## 4.3 QUANTITATIVE RESULTS

Tab. 1 shows the performance of different planners on the Navtest Benchmark with PDM score. We see that: i) Neural planners score low on Drivable Area Compliance (DAC) as the aforementioned methods are unable to accurately determine the extent of the drivable area, leading to potential misclassification of road boundaries and off-road regions. ii) Our method eliminates the use of lidar inputs, relying solely on image data and a lightweight ResNet34 backbone with fewer parameters, while still achieving state-of-the-art performance. Notably, the Drivable Area Compliance (DAC) score improved by 2.9% over the previous best method. Overall, the PDM score increased by 1.1%, representing a significant advancement in navigation accuracy. iii) We scale up the image backbone using the V2-99 (Lee et al., 2019) architecture, and observe that with a larger backbone, our method surpasses the rule-based teacher PDM-Planner by 1.9% and improves upon the ResNet-based backbone by 3.4%. In contrast to the findings in Hu et al. (2023), which suggests that larger backbones yield only minor enhancements in planning performance, our results show a particularly notable improvement in the EP, LK, and DAC metrics. This underscores the significant scalability of our approach when utilizing a larger backbone.

Tab. 2 illustrates the performance of various planners on the Navtest Benchmark with an Extended PDM Score. As illustrated in Tab. 1, the identical pattern is evident. Moreover, the method continues to perform exceptionally well on the new metrics, particularly in Extended Comfort. This indicates that the vehicle rarely exhibits inconsistencies in its behaviours over time. Furthermore, it is evident that the incorporation of the pre-designed rule-based teacher (DDC, LK, and EC) during distillation has a negligible effect on the original metrics, namely NC, DAC, EP, TTC, and C.

## 4.4 ABLATION STUDY

Tab. 3 and Tab. 4 illustrate the results of the ablation study on various modules employed in our network. **W** employs weighted confidence during inference, as discussed in Sec. 3.3. **TS** integrates the Temporal SE module, while **P** utilizes extra perception tasks for auxiliary supervision (Chitta et al.,

| W | TS | P | Backbone | NC ↑ | DAC ↑ | EP ↑ | TTC ↑ | C ↑ | PDMS ↑ |
|---|----|---|----------|------|-------|------|-------|-----|--------|
| - | - | - | Resnet34 | 97.5 | 92.0 | 80.4 | 91.8 | 100 | 85.0 |
| ✓ | - | - | Resnet34 | 97.4 | 96.0 | **81.0** | 92.8 | 100 | 86.5 |
| ✓ | ✓ | - | Resnet34 | **97.6** | **96.0** | 80.4 | 93.1 | **100** | **86.6** |
| ✓ | ✓ | ✓ | Resnet34 | 97.6 | 95.6 | 80.1 | **93.3** | 100 | 86.1 |

Table 3: **Ablation study on the Navtest Benchmark with original metrics.** W: Weighted confidence during inference. TS: Temporal SE module in perception network. P: Perception tasks are used for auxiliary supervision.

| W | TS | P | Backbone | NC ↑ | DAC ↑ | EP ↑ | TTC ↑ | C ↑ | TL ↑ | DDC ↑ | LK ↑ | EC ↑ | EPDMS ↑ |
|---|----|---|----------|------|-------|------|-------|-----|------|-------|------|------|---------|
| - | - | - | Resnet34 | 97.7 | 92.1 | **80.5** | 92.6 | 100 | 100 | 98.1 | 67.0 | 92.3 | 76.8 |
| ✓ | - | - | Resnet34 | 97.9 | 96.5 | 80.3 | **93.7** | 100 | 100 | 98.8 | 66.2 | 92.3 | 79.8 |
| ✓ | ✓ | - | Resnet34 | **97.9** | **96.5** | 79.2 | 93.4 | **100** | **100** | **98.9** | **67.2** | **97.7** | **80.6** |
| ✓ | ✓ | ✓ | Resnet34 | 97.9 | 96.2 | 78.6 | 93.4 | 100 | 100 | 98.6 | 67.1 | 97.5 | 80.3 |

Table 4: **Ablation study on the Navtest Benchmark with extended metrics.** W: Weighted confidence during inference. TS: Temporal SE module in perception network. P: Perception tasks are used for auxiliary supervision.

2022). We observe that the weighted confidence leads to an enhanced PDM Score and Extended PDM Score, which suggests that weighted confidence during inference is a crucial step. Intuitively, the movement of a vehicle is more susceptible to collisions and drivable areas, and therefore the weights need to be larger in these contexts. The inclusion of Temporal SE further improves the score, particularly the Extended Comfort (EC) subscore, which rises from 92.3 to 97.7. This underscores its effectiveness in capturing temporal features that enhance overall smoothness of model predictions. Additionally, we find that auxiliary training of perception worsens the planning performance, suggesting that perception does not positively impact the performance within our framework.

| Method | Backbone | NC ↑ | DAC ↑ | EP ↑ | TTC ↑ | C ↑ | TL ↑ | DDC ↑ | LK ↑ | EC ↑ | EPDMS ↑ |
|--------|----------|------|-------|------|-------|-----|------|-------|------|------|---------|
| Hydra-MDP++ | ResNet34 | 97.6 | 96.0 | 80.4 | 93.1 | 100 | 99.9 | 97.5 | 65.5 | 97.4 | 79.5 |
| Hydra-MDP++* | ResNet34 | 97.9 | 96.5 | 79.2 | 93.4 | 100 | 100 | 98.9 | 67.2 | 97.7 | 80.6 |
| Hydra-MDP++ | V2-99 | 98.6 | 98.6 | 85.7 | 95.1 | 100 | 100 | 97.8 | 67.6 | 96.7 | 83.4 |
| Hydra-MDP++* | V2-99 | 98.8 | 97.8 | 84.0 | 95.3 | 100 | 100 | 99.1 | 70.1 | 96.8 | 84.1 |

Table 5: **Ablation Study on the Navtest Benchmark with extended metrics.** *Extended metrics act as rule-based teachers during Hydra-Distillation.

Tab. 5 presents an ablation analysis of distillation with extended rule-based teachers. Models marked with * indicate that the new metrics were incorporated as additional distillation targets. As shown, Hydra-MDP++ * achieved significant improvements in metrics like Driving Direction Compliance (DDC), Lane Keeping (LK), and Extended Comfort (EC), confirming the successful integration of rule-based knowledge from the new teachers. Furthermore, the considerable increase in the Extended PDM Score (EPDMS) highlights the overall advantages of adding these metrics, reflecting better alignment between model predictions and ideal driving behaviors. This improvement not only enhances rule-compliance but also provides a more robust framework for safe, human-like driving.

## 4.5 QUALITATIVE RESULTS.

In Fig. 3, three representative driving scenarios are displayed, showcasing how our Hydra-MDP++ model performs end-to-end trajectory planning. The left-hand side compares the ground truth trajectory (green) and the planned trajectory (red). The top image shows a right turn where the model accurately follows the curve. The middle image displays straight driving in a dense urban environment, maintaining a safe distance from other vehicles. The bottom image depicts the car coming to a stop at a red light, demonstrating proper deceleration and safety compliance.

On the right-hand side, we offer the evaluation of 8192 candidate trajectories scored across five metrics: NC (No at-fault Collision), DAC (Drivable Area Compliance), TTC (Time-to-Collision), EP (Ego Progress), and LK (Lane Keeping), with EPDMS being an aggregated score. The visualization reveals the distribution of these scores, with higher scores indicating trajectories that balance both safety and progress. These metrics are essential for the planning process of Hydra-MDP++ ,

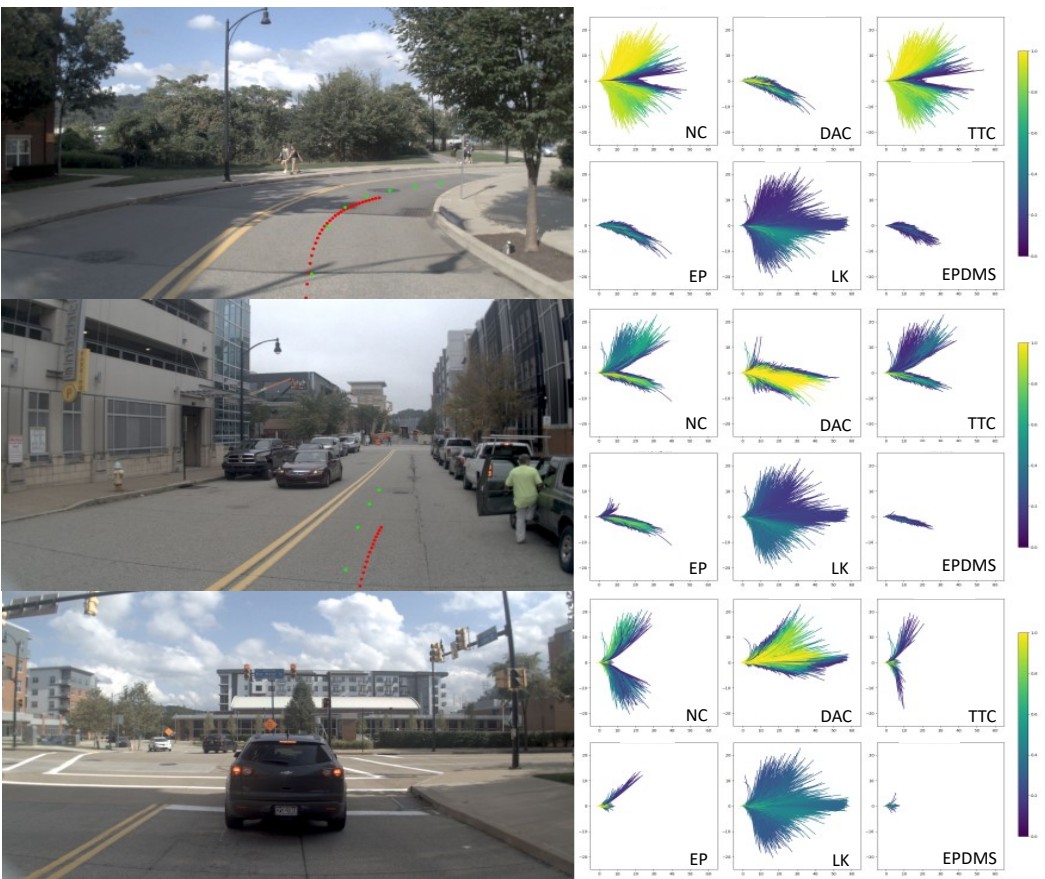

Figure 3: Visualizations of our planned trajectory (red dots), ground-truth trajectory (green dots), and predicted scores for different metrics. Trajectories scoring less than 0.1 are omitted.

ensuring it selects optimal trajectories that align with both safety constraints and efficient driving behaviors. The color gradients represent the evaluation of the candidate trajectories based on different metrics. Lighter colors (e.g. yellow or green) correspond to higher scores, indicating more optimal trajectories according to the specific metric. In contrast, darker colors (e.g. purple or blue) correspond to lower scores, representing less favorable trajectories.

## 5 CONCLUSION

We present Hydra-MDP++ , a state-of-the-art end-to-end motion planner designed to synergize the strengths of rule-based and neural planning methodologies. By learning from extensive human driving demonstrations and the insights provided by rule-based experts, Hydra-MDP++ can navigate complex environments more effectively. Our expert-guided hydra-distillation paradigm aligns the planner with simulation-based metrics in NAVSIM, enhancing its reliability and performance. To address the shortcomings of existing evaluation metrics, we have expanded the teacher model to include crucial aspects such as Traffic Lights Compliance, Lane Keeping Ability, and Extended Comfort. This comprehensive approach ensures that the decision-making process in driving scenarios is robust, adaptable, and adheres to safety standards. The evaluation of Hydra-MDP++ using the public NAVSIM dataset reveals its superior performance compared to traditional methods, demonstrating that it achieves high levels of accuracy and compliance with fewer parameters, which highlights its efficiency and effectiveness in real-world applications. This advancement not only improves driving performance but also contributes to the ongoing development of safer and more intelligent autonomous vehicles.

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
