# OpenReview forum: "Hydra-MDP++: Advancing End-to-End Driving via Hydra-Distillation with Expert-Guided Decision Analysis"
_ICLR.cc/2025/Conference — ICLR 2025 Conference Withdrawn Submission_

### Official Review · Reviewer_LUMZ · 2024-10-24

**Soundness:** 2
**Presentation:** 3
**Contribution:** 1
**Rating:** 3
**Confidence:** 5

**Summary:**

This paper proposes to complement imitation learning with an extra term that punish or reward different trajectories based on their rollouts.

**Strengths:**

1. The emperical result is strong.
2. The proposed new metrics are a good update for NAVSIM.
3. The paper is easy to follow.

**Weaknesses:**

Limited novelty: the contributions of this work is on the decoder/planner side, which is actually not quite related to end-to-end autonomous driving.  In pure planning side, **it is not new to conduct rollout during training and then reward/punish the planner like in [1, 2]**.

[1] Closing the Planning-Learning Loop with Application to Autonomous Driving. TRO
[2] Rethinking Imitation-based Planners for Autonomous Driving. ICRA 2024

In summary, it seems that this work simply adopts the long existing idea in the planning field: offline rl by rollout during training into a new benchmark NAVSIM.  As a result, I do not think implementing an image encoder with existing decoder designs have enough contributions and excitement to reach the bar of ICLR. Thus, I give reject rating. If the authors could demonstrate advantages over planning baselines like [1,2] in planning benchmarks like nuPlan,  I am glad to improve my rating.

**Questions:**

None

---

### Official Review · Reviewer_B2Ks · 2024-11-03

**Soundness:** 3
**Presentation:** 2
**Contribution:** 2
**Rating:** 5
**Confidence:** 4

**Summary:**

The paper proposes a a motion prediction driving approach for driving agents
 which fuses a direct error prediction loss with simulation metrics based on NAVSIM.
The paper shows promising results on the NAVSIM benchmark.

**Strengths:**

The paper goes into an important direction which is incorporating actual driving metrics into end-to-end driving learning instead of pure action or target point prediction. This is probably more relevant and translate better to actual driving quality results.

**Weaknesses:**

* The main weakness I observed is that I failed to understand the explanation of this as a hybrid method where there would be some additional framework  to include IDM type of driving with a learned method. What I understood is the use of NAVSIM simulation based metrics which adds closed loop type of metrics into the loss function giving a bit more driving rollout supervision. I guess this is arguably similar to a hybrid approach because the rule base in embedded in the NAVSIM. Still that is not fully clear.
I think the idea is simple and effective but is masked with a description that is bigger and more complex than necessary.

*  The results are promising, but I would guess that optimizing for NAVSIM simulation metrics would lead for better navsim results. I would be more confident in using this learning strategy is full closed loop benchmarks where used with other platforms like CARLA or some other realistic simulator.

**Questions:**

I wonder what are the potential drawbacks on using simulation based targets like navsim in terms of compute cost for training as that should be considerable more expensive than using datasets directly for the evaluation.

What are also the impacts on overall determinism when using simulation as a part of the training framework. That should increase the training complexity.

---

### Official Review · Reviewer_TbsM · 2024-11-04

**Soundness:** 2
**Presentation:** 3
**Contribution:** 2
**Rating:** 5
**Confidence:** 4

**Summary:**

This paper presents Hydra-MDP++, an extension of existing work Hydra-MDP, which could give decisions learned from rule-based experts and human demonstrations. Following the framework proposed in Hydra-MDP, Hydra-MDP++ achieves state-of-the-art performance on NAVSIM benchmark by receiving feedback from multiple expert teachers to help its trajectory decoder integrate the knowledge of rule-based and neural planners. Moreover, Hydra-MDP++ leverages a temporal Squeeze-and-Excitation network for temporal feature fusion from tiny image encoders, and expands teacher models to include more human knowledge.

**Strengths:**

1. The writing is easy to follow.
2. The idea of leveraging both simulation feedback and human demonstrations as supervision is interesting and intuitive.
3. Clear illustration of the framework architecture. (Fig. 2).

**Weaknesses:**

1. Missing Hydra-MDP baseline in experiments (Table 1 and Table 2). As the proposed method Hydra-MDP++ is an extension of Hydra-MDP, it is important to compare their results under the same evaluation protocol to demonstrate performance improvement. However, the results of the original Hydra-MDP are missing in the main experiments.
2. Lack of discussion on the difference between Hydra-MDP and Hydra-MDP++. The authors did mention several modifications throughout the paper such as in Sec. 3.4. However, it's still less obvious how Hydra-MDP++ differs from Hydra-MDP as a whole and what design choices are made to improve the original Hydra-MDP. A detailed section to elaborate on their differences and the motivation behind these modifications should be included. Without such clarification, I can hardly tell the novelty behind this paper.
3. A potential architectural flaw. The key innovation of this work is to distill the simulation feedback in the training stage, and then use the predicted feedback as guidance to select the best action to execute. However, a notable limitation of this design would be the over-reliance on a pre-developed driving simulator, which does not exist for most driving datasets. Therefore, the application of this method might be limited to certain datasets with the paired simulator. It's encouraged to include this drawback in the discussion or limitation section.
4. Missing details of inference latency which is critical for autonomous driving models. As Hydra-MDP++ is a sampling-based planning method, the planner estimates multiple action proposals and finally executes the most plausible one, it is important to report the latency/speed of such multi-round estimation. Also, the inference speed of the whole proposed pipeline should be included.

**Questions:**

Please see the weakness section above.

---

### Official Review · Reviewer_77mX · 2024-11-05

**Soundness:** 3
**Presentation:** 2
**Contribution:** 3
**Rating:** 6
**Confidence:** 3

**Summary:**

This paper introduces Hydra-MDP++, an end-to-end autonomous driving framework that bridges rule-based and neural planners by incorporating human demonstrations and expert guidance. Using a teacher-student knowledge distillation model, Hydra-MDP++ enables a multi-head student decoder to learn and validate trajectory proposals across various aspects of safe driving. Tested on the NAVSIM benchmark, Hydra-MDP++ shows impressive performance, achieving high compliance with metrics like lane-keeping, comfort, and traffic light observance. The framework demonstrates the potential for more adaptive and resilient autonomous driving by balancing human-like flexibility with rule-based reliability.

**Strengths:**

1. The integration of rule-based and neural planning is innovative. The paper’s approach effectively combines rule-based and neural planners, adding interpretability and adaptability to neural models, which may benefit real-world applications.
2. By incorporating traffic light compliance, lane-keeping ability, and extended comfort, the model considers crucial safety and regulatory aspects, showing a clear improvement over existing methods on the NAVSIM benchmark.
3. The model achieves high accuracy and compliance using only a ResNet-34 backbone, highlighting the computational efficiency.

**Weaknesses:**

For me, I have no main concerns of this paper. Here are some personal questions.
1. The specifics of the multi-head decoder’s feedback integration could be further clarified. While it’s stated that each head represents a different safety component, it would be beneficial to describe how conflicts between heads are resolved.
2. The qualitative examples primarily showcase straightforward scenarios like lane-following and traffic light compliance. Could authors also show some including challenging cases, such as complex intersections or mixed traffic environments?
3. I am not sure if this method can be evaluated on more than one dataset to prove generalization abilities.

**Questions:**

Here is one question: The model performs well on NAVSIM, yet it would be beneficial to discuss its scalability to other benchmarks or real-world tests.

---

### Note · Authors · 2024-11-13

I have read and agree with the venue's withdrawal policy on behalf of myself and my co-authors.